



# Necessary conditions for algorithmic tuning of weather prediction models using OpenIFS as an example

Lauri Tuppi[1], Pirkka Ollinaho[2], Madeleine Ekblom[1], Vladimir Shemyakin[3], and Heikki Järvinen[1]

[1]Institute for Atmospheric and Earth System Research / Physics, University of Helsinki, Finland
[2]Finnish Meteorological Institute, Helsinki, Finland
[3]School of Engineering Science, Lappeenranta University of Technology, Lappeenranta, Finland

**Correspondence:** Lauri Tuppi (lauri.tuppi@helsinki.fi)

**Abstract.** Algorithmic model tuning is a promising approach to yield the best possible forecast performance of multi-scale multi-phase atmospheric models once the model structure is fixed. The problem is to what degree we can trust algorithmic model tuning. We approach the problem by studying the convergence of this process in a semi-realistic case. Let $M(\boldsymbol{x}, \boldsymbol{\theta})$ denote the time evolution model, where $\boldsymbol{x}$ and $\boldsymbol{\theta}$ are the initial state and the default model parameter vectors, respectively. A

*necessary condition* for an algorithmic tuning process to converge is that $\boldsymbol{\theta}$ is recovered when the tuning process is initialised with perturbed model parameters $\boldsymbol{\theta}'$ and the default model forecasts are used as pseudo-observations. The aim here is to gauge which conditions are *sufficient* in semi-realistic test setting to obtain reliable results, and thus building confidence on the tuning in fully-realistic cases. A large set of *convergence tests* is carried in semi-realistic cases applying two different ensemble-based parameter estimation methods and the OpenIFS model. The results are interpreted as general guidance for algorithmic

model tuning, which we successfully tested in a more demanding case of simultaneous estimation of eight OpenIFS model parameters.

## 1 Introduction

Numerical weather prediction (NWP) models solve non-linear partial differential equations in discrete and finite representation. Sub-grid scale physical processes, such as cloud micro-physics, are treated in specific closure schemes. Once the model

structure is fixed, some parametric uncertainty remains due to the fact that the closure parameter values are not known exactly. This uncertainty can be conceived as a probability density of the closure parameters: the expected value corresponds to the optimal model skill and the co-variance to their inherent uncertainty. The expected parameter value is the obvious choice for deterministic forecasting while the co-variance can be utilized to represent parametric uncertainty in ensemble forecasting (Ollinaho et al., 2017).

Model tuning is an attempt to unveil some statistics of the probability density of the closure parameters, and algorithmic methods add objectivity and transparency to the process (Hourdin et al., 2017; Mauritsen et al., 2012). It is paramount in algorithmic model tuning that the method applied is able to converge to the correct statistics with limited computing resources.

The aim of this paper is to gauge the circumstances favouring successful model tuning when ensemble-based parameter estimation methods are used. These include, for instance, differential evolution (Storn and Price, 1997) and its variants, genetic





algorithm (Goldberg, 1989), particle swarm optimisation (Kennedy, 2010) and Gaussian importance samplers (e.g. EPPES, Järvinen et al., 2012; Laine et al., 2012). The results may also be useful for more deterministic algorithms, such as multiple very fast simulated annealing (Ingber, 1989), and filter based estimation algorithms such as ensemble Kalman filter and its variants (e.g. Annan et al., 2005; Pulido et al., 2018) as well as particle filters (e.g. Kivman, 2003). Of course, the length of the time window is limited in filter based estimation methods.

The results are interpreted so as to provide guidance into successful tuning exercises and savings in computing time. Convergence testing is always semi-realistic and can provide insight how to design fully-realistic model tuning exercises. Based on a very large amount of testing, the following synopsis guidance can be drawn up:

1. The level of realism (synopsis of our guidance: default model must be recovered in trivial testing when model parameters are off-set slightly; initial state perturbations tend to enhance convergence properties)

2. The optimisation target (too simplistic target function may not allow convergence even in trivial testing, or convergence is very slow; convergence benefits of comprehensive target function formulation)

3. Efficient use of computations (long convergence testing with a small ensemble and varying initial states is better than a short test with a large ensemble; in OpenIFS, 24 hour forecast range seems to be long enough for good parameter sensitivity and convergence)

4. Reproducibility of results (24 hour forecast range seems optimal; at longer ranges than 24 hours, results are less repeatable)

5. Can one trust on algorithmic tuning (not blindly, take it as expert-guided)

6. Potential pitfalls (initial parameter off-set should not be too large, such as off by factor of two; initial parameter off-set and uncertainty should be proportional; be aware that optimal parameter values can be dependent on the forecast range

applied)

## 2 Methods

### 2.1 OpenIFS and closure parameters of the convection scheme

OpenIFS is the atmospheric forecast model of the Integrated Forecasting System (IFS) of ECMWF. In this study, we use a version based on the model version being operational from November 2013 to May 2015 (cycle 40r1). Convergence tests are

run at two resolutions: TL159 and TL399 corresponding to 125 km and 50 km resolutions, respectively, both with 91 vertical levels. Initial conditions are extracted from the ECMWF operational archive (a control member plus 50 perturbed analyses) for year 2017 to cover different weather regimes and seasons. Each convergence test contains 52 ensemble forecasts, i.e., an ensemble is initialised once per week. The data set of ensemble initial conditions (Ollinaho et al., in prep) has been generated with the IFS cycle 43r3. Thus, some spinup/spindown is possible at early forecast ranges of a few hours.





We focus on closure parameters of the convection scheme, consisting of a bulk mass flux with an updraught and downdraught pair in each grid box for shallow, deep and mid-level convection (ECMWF, 2014; Bechtold et al., 2008). The parameters, their default values and short descriptions are in Table 1. The tests also involve the use of a stochastic representation of model uncertainty (Stochastically Perturbed Parametrisation Tendencies, (SPPT, Palmer et al., 2009)).

## 2.2    OpenEPS – ensemble prediction workflow manager

Convergence tests involve running large amounts of ensemble forecasts. Traditionally, ensemble forecasting and research on ensemble methods have been tied to major NWP centres providing operational ensemble forecasts to end-users. Usually these platforms are not suited for academic research. Instead, we use a novel and easily portable ensemble prediction workflow manager (called OpenEPS), developed at the Finnish Meteorological Institute specifically for academic research purposes (https://github.com/pirkkao/OpenEPS).

OpenEPS has been designed for launching, running, and post-processing a large number of ensemble forecast experiments with only a small amount of manual work. OpenEPS is very flexible and can be easily coupled with external applications required in parameter tuning, such as autonomous parameter sampling.

## 2.3    Optimisation algorithms

Brute force sampling of the parameter space of full-complexity NWP models is computationally far too expensive. Typically,
one can afford running perhaps only some tens or a maximum of a few hundred simulations in a tuning experiment. Therefore, the tuning methods need to be sophisticated. In these convergence tests, we use two ensemble based optimisation algorithms: Ensemble prediction and parameter estimation system (EPPES) (Laine et al., 2012; Järvinen et al., 2012) and differential evolution (DE) (Storn and Price, 1997; Shemyakin and Haario, 2018).

    EPPES is a hierarchical statistical algorithm, which uses Gaussian proposal distributions, importance sampling, and sequen-
tial modelling of parameter uncertainties to estimate model parameters. A parameter sample is drawn from the distribution, an ensemble forecast is run with these parameter values and goodness of the parameter values is evaluated by calculating a cost function for each ensemble member. The proposal distribution is sequentially updated such that it shifts towards more favourable parameter values. Here, the shift of parameter mean values between consecutive iterations is limited to a conservative value of 5%.

DE (Storn and Price, 1997) is heuristically based on natural selection. It consists of evolving population of parameter vectors where vectors leading to good cost function values thrive, and produce an offspring, while vectors leading to bad cost function values are eliminated. The population update procedure of DE is achieved by a certain combination of mutation and crossover steps. These steps ensure that the new parameter vectors differ at least slightly from the vectors that already belong to the population. The natural selection is achieved through the selection step, where the elements of the current population compete
with the new candidates based on the defined cost function. The fittest are kept alive and proceed to the next generation. Besides the standard DE algorithm, we use: generation jump (Chakraborty, 2008), DE/best/1 mutation strategy (Feoktistov, 2006; Chakraborty, 2008; Qing, 2009) with randomized scale factor by jitter and dither (Feoktistov, 2006; Chakraborty, 2008)





and recalculation step (Shemyakin and Haario, 2018). Recalculation step has been designed to enhance convergence when the cost function is stochastic. Occasionally truly good parameter values lead to bad cost function values due to other perturbations in the forecast. However, the recalculation step passes the previous population for a new iteration, which is used to confirm that those parameter vectors, which lead to good scores on previous iteration, lead to good cost function values again.

The main focus here is on the EPPES results. More details about the algorithms and their specific setting are explained in Appendices 1 and 2.

### 2.4 Optimisation target functions

We will apply two very different optimisation target functions (hereafter cost functions) in the convergence tests. The first one is the global root mean squared error of the 850 hPa geopotential height ($\Delta Z$):

$$\Delta Z = \sqrt{\frac{1}{D} \int_{D} (Z_{850} - Z'_{850})^2 dD} \tag{1}$$

where $Z_{850}$ and $Z'_{850}$ denote 850 hPa geopotential in the control forecast and perturbed forecast at each grid point. $D$ denotes the horizontal domain. This is a restrictive cost function formulation and used here merely as a useful demonstrator. There are three reasons why we expect $\Delta Z$ to perform sub-optimally. Firstly, it exploits information only from a small fraction of the model domain and the upper parts of the model domain remain unconstrained. Secondly, it requires substantial interpolation of forecasts and reference data. Finally, the 850 hPa level intersects with the ground level in mountainous areas.

The second one is the global moist total energy norm ($\Delta E_m$, e.g. Ehrendorfer et al., 1999) and it is a very comprehensive integral measure of the distance between two atmospheric states. The moist total energy norm can be written as

$$\Delta E_m = \frac{1}{2} \int_{\eta} \int_{D} [u'^2 + v'^2 + c_q \frac{L^2}{c_p T_r} q'^2] dD \frac{\delta p_r}{\delta \eta} d\eta + \frac{1}{2} \int_{D} [R \frac{T_r}{p_r} \ln p_s'^2] dD, \tag{2}$$

where $u'$, $v'$, $T'$, $q'$ and $\ln p_s'$ refer to differences between forecast and pseudo-observations in wind components, temperature, specific humidity and the logarithm of surface pressure. $c_q$ is a scaling constant for the moist term, $L$ vaporisation energy of water, $c_p$ specific heat at constant pressure, $T_r$ reference temperature and $p_r$ reference pressure. Here, we set $T_r = 280$ K and $p_r = 1000$ hPa as in (Ollinaho et al., 2014). $D$ and $\eta$ denote increments of horizontal and vertical integrals. Unlike in Ollinaho et al. (2014) we set $c_q$ to 1 and $d\eta$ to equal the difference of pressure between consecutive model levels. Ollinaho et al. (2014) also show instructions how to discretise $\Delta E_m$ for practical use.

We expect that at short forecast ranges, the linkage between variations in the values of $\Delta E_m$ and parameter perturbations is detectable, enabling to estimate parameter densities.

### 2.5 Evaluation of convergence tests

The parameter convergence is measured with fair continuous ranked probability score (fCRPS, Ferro et al., 2008) formulated as the kernel representation (see e.g. Leutbecher, 2018). fCRPS rescales the scores as if the ensembles were infinitely large so





that there is no dependence between ensemble size and the score itself. The property of fairness is essential in comparison of convergence tests with different ensemble sizes. fCRPS has not been designed for evaluation of ensembles of parameter values, so a direct application of fCRPS may lead to cancellation of the two terms (see Eq. 6, Leutbecher, 2018) causing difficulties in interpreting the results. Therefore, we use the two terms separately for each parameter $\theta_n$:

$$\text{fCRPS}_1 = \frac{1}{M} \sum_{j=1}^{M} |\theta'_{j,n} - \theta_{d,n}| \tag{3}$$

and

$$\text{fCRPS}_2 = \frac{1}{2M(M-1)} \sum_{j=1}^{M} \sum_{k=1}^{M} |\theta'_{j,n} - \theta'_{k,n}|, \tag{4}$$

where $\theta'_{j,n}$ and $\theta'_{k,n}$ are the parameter values used by ensemble members $j$ and $k$, $\theta_{d,n}$ is the default parameter value and $M$ is the ensemble size. The first part is a measure of how much the ensemble mean parameter value differs from the default value while the second part indicates how much spread is associated with the ensemble mean parameter value; in other words how certain or uncertain the algorithm considers the parameter value. The perfect score of each part of fCRPS is zero.

## 3 Set-ups of the convergence tests

The outline of our experiment set-ups is shown in Table 2. The table explains the different levels of realism we use in the convergence tests. The different levels of realism are tested to see which is the optimal way to extract information of the parameter space. On the one hand, keeping convergence tests as simple as possible makes interpretation of the test results easy, but on the other hand, more realistic tests provide information on how the parameter tuning system would perform in fully-realistic tuning tasks.

L0, L1 and L3 tests are performed with one set-up: a forecast range of 48 hours and an ensemble size of 50 members. $\Delta Z$ is only tested at L1 level. Most of the effort is put on L2 testing with $\Delta E_m$ so they are performed with various combinations of forecast range and ensemble size. The focus is on L2 tests on one hand because we assume that if the convergence is good at this level of realism, it will be good also at lower levels. On the other hand, convergence in L2 and L3 tests is relatively similar. We use the parameters $\theta_1$ and $\theta_2$ in L2 tests and parameters $\theta_1$ to $\theta_5$ in L3 tests. In L0 tests all forecasts are initialised from an unperturbed initial state of the control forecast (1 Jan 2017 00UTC). L1, L2 and L3 tests use ensemble initial conditions from 52 dates.

## 4 Results

For brevity, we mainly concentrate on discussing results obtained with EPPES, although most of the convergence tests have been run with both algorithms. Due to the nature of the algorithms, EPPES produces less noise near the end of the convergence





tests. Therefore, results generated with EPPES are easier to interpret. However, none of the results produced with DE contradict
the results of EPPES.

## 4.1   Selection of level of realism

We test how much realism should be included in algorithmic tuning. Figure 1 shows four convergence tests with different levels
of realism, described in Table 2. As expected the parameters converge slower the higher the level of realism is as the parameter
uncertainties decrease slower. Both parameters converge very fast in the trivial L0 convergence test. Convergence to the default
values in the L0 convergence test is trivial as minimisation of the parameter perturbations is the only way to minimise the
cost function values. However, we want to emphasize that fully-realistic tuning at L0 level of realism could lead to unwanted
results since the parameters would be optimised for that specific weather state only. L0 can still be used to test that the tuning
infrastructure works.

L1, L2 and L3 tests resemble more fully realistic model tuning. Figure 1 shows that the convergence tests at different levels
of realism behave quite reasonably. The uncertainty cannot vanish completely since some uncertainty is always present due to
ensemble initial conditions. The only difference between L1, L2 and L3 tests is that more complex tests converge slower.

L1, L2 and L3 tests have a common feature that the parameters tend to converge to some off-default values. This feature is
inherent to convergence tests, which use pseudo-observations generated with the control model. The convergence to off-default
values will be discussed in further below.

We recommend using L1 (only perturbed initial conditions) in fully-realistic model tuning. L1 is the simplest safe option.
Higher levels of realism do not provide additional information, instead, they only make convergence more difficult. L0 is only
recommended for testing purposes. Depending on user needs, L1 can be modified so that more parameters are added.

## 4.2   Selection of optimisation target

Here, we compare convergence tests that use different cost functions. Figure 2 compares L1 convergence tests with $\Delta E_m$
(shown with black solid and dash-dotted lines) and $\Delta Z$ (shown with cyan dotted line and shading in the background). Figure
2 shows that $\Delta E_m$ leads to much faster convergence than $\Delta Z$. The superior performance of $\Delta E_m$ is explained by: firstly,
global integral of several variables catches the signal of parameter perturbations much better than a single level measure.
Secondly, perturbations of convection parameters do not affect 850 hPa geopotential directly so 48 hours may not be long
enough to develop a traceable signal. Perturbations of convection parameters modify specific humidity, wind components and
170  temperature directly. These fields must change substantially before the signal is seen in geopotential at 850 hPa.

The results showed that a more comprehensive cost function means faster and more reliable convergence. Therefore, we
recommend using such a measure that accounts for more than one atmospheric level and more than one variable at least in case
of convection parameters. However, it is possible that with some other parameters a cost function other than $\Delta E_m$ might work
better.



### 4.3 Finding the most efficient set-up

This subsection aims at finding a forecast range and an ensemble size that give satisfactory convergence with a minimal amount of computational resources. The consideration is taken at two steps: first, we compare convergence tests that use different forecast ranges, and then we concentrate on the forecast range with the fastest convergence in order to find optimal ensemble size. Using L2 level of realism. We expect that the results obtained with such a high level of realism generalise well to lower levels.

Figure 3 shows the components of fCRPS of the final ensemble for the parameter $\theta_2$ from numerous convergence tests with different forecast ranges and ensemble sizes. From Figure 3 it is obvious that the convergence of $\theta_2$ is the most efficient when the forecast range is 24 hours. It is remarkable that 24 hours and also 48 hours lead to significantly better convergence than 72 hours that was used for example in (Ollinaho et al., 2014). For $\theta_1$, 12, 24 and 48 hour forecasts are the best and roughly equally good (not shown).

The superior convergence with 24 hour forecasts can be explained by relatively linear response of OpenIFS to parameter perturbations, which $\Delta E_m$ is able to detect. The sub-optimal performance of 12 hour forecasts compared to 24 hour forecasts may be, at least partly, due to the spin-up related to to discrepancy of model versions. Consequently, we are somewhat uncertain about the true performance at very short forecast ranges. Against our expectations, some convergence occurs also at the longest forecast ranges when the response to parameter perturbations is definitely non-linear. At least some parameter convergence takes place with all forecast ranges but the convergence is by far the fastest at short ranges.

Figure also 3 shows that there is relatively more error in the parameter mean value than there is spread when long or very short forecasts are used. This is discussed further below.

Figure 4 concentrates on the forecast range of 24 hours, and shows the evolution of convergence tests with different ensemble sizes again measured with fCRPS. Firstly, Figure 4 indicates that convergence tests with large enough ensemble size are stable. Three convergence tests having the smallest ensemble sizes do not show the desired smooth decrease of both parts of the fCRPS. Sampling variance seems to have a strong effect in those cases. Sampling variance seems to play a smaller role when the ensemble size is 20 or larger. Secondly, Figure 4 enables comparison of the convergence tests from the resources point of view. For example, tests with 50 ensemble members and 20 iterations, and 20 members and 50 iterations both use 1000 forecasts. However, the latter option leads to much better convergence. The same pattern seems to apply to most of the similar pairs. Increasing the ensemble size beyond ∼20 members does not seem to be justified by the computational resources point of view. Figure 4 shows the results only for $\theta_2$ but the same applies also to $\theta_1$ although the results with $\theta_1$ are less conclusive (not shown). Interestingly, these results are in line with the conclusions of Leutbecher (2018) that in ensemble forecasting related research it is better to have large number of small ensembles than small number of large ensembles.

Based on these results we recommend using relatively short forecasts of 24 hours, at least when convection parameters are concerned. We also recommend using medium-sized ensembles of about 20 members. Very small ensembles of less than 10 members increase sampling variance and destabilise the convergence. Moreover, convergence of DE was practically impossible





with small ensembles. We are fairly sure that ∼20 members is close to optimal ensemble size at least for tuning these two parameters. However, we are somewhat uncertain that 24 hours is optimal forecast range for all parameters.

## 4.4  Reliability of convergence tests

This subsection shows additional evidence to use L1 level of realism and 24 hour forecasts. Two example convergence tests are repeated four times first using L1 and then L2: first with a sub-optimal set-up of 48 hour forecasts and 20 ensemble members and second with a close to optimal set-up of 24 hour forecasts and 26 ensemble members. These set-ups are highlighted in Figure 3. We test whether these convergence tests yield similar results every time i.e. the repeatability.

Figure 5 shows the evolution of the repeated convergence tests measured with fCRPS in a similar fashion as in Figure 4. L1 convergence tests are on the left-hand side and L2 tests on the right-hand side. Labels A1 to A4 refer to the sub-optimal set-up and labels B1 to B4 to the optimal set-up. The left-hand side of Figure 5 shows that both set-ups yield fairly reproducible convergence at L1. However, when the level of realism is raised to L2, only the more optimal set-up seems to yield repeatable convergence with EPPES. The results obtained with DE are less conclusive as DE tends to fluctuate around the optimum (not shown).

We recommend using such a set-up that is the most likely to yield reliable parameter convergence. At least in our case, the optimal set-up of 24 hour forecasts and ∼20 member ensembles is also the most reliable set-up. Also, using only initial condition perturbations (L1) besides the parameter perturbations leads to more reliable convergence than initial condition plus stochastic model perturbations (L2).

## 4.5  Potential pitfalls

Firstly, we noticed that some parameters tend to converge to some off-default values. As an example, the two most used parameters ($\theta_1$ and $\theta_2$) tended to converge to slightly different values depending on the forecast range used. $\theta_2$ tends to converge to a value smaller (larger) value than the default value when forecasts longer (shorter) than 24 hours are used. In this respect $\theta_1$ behaves like a mirror-image to $\theta_2$. However, at least $\theta_2$ tends to converge in one parameter convergence tests in the similar way as in the two parameter tests. This is illustrated in Figure 6, which shows the mean values of the final parameter for $\theta_1$ (to the left) and for $\theta_2$ (to the right). Especially convergence of $\theta_2$ seems to depend very strongly on the forecast range used in the convergence tests. We tested whether it is actually possible to obtain smaller cost function values by running regular ensemble forecasts with the parameter values proposed by the algorithm (see Figure 6) and regular ensemble forecasts with default parameter values. Both sets of ensemble forecasts were compared to respective control forecasts with $\Delta E_m$. The tests were repeated with only initial condition perturbations active and with initial condition plus stochastic model perturbations active. The results indicated that the globally optimal parameter values and the parameter default values do not necessarily match even though the control model is used as reference. It is indeed possible to obtain lower cost function values with some off-default parameter values than with the default parameter values. This means that the peculiar dependence is not caused by any deficiencies in the cost function or optimisation algorithms. However, convergence tests and fully-realistic tuning are





so different that we are unsure whether this dependency even exists at all in fully-realistic tuning, and if it exists, does the dependency actually hinder tuning after all.

A potential pitfall might emerge if there is a need to do algorithmic tuning with ensembles of very different sizes. At least the two algorithms, EPPES and DE, are difficult to set up so that they would work satisfactorily regardless of the ensemble size. If the algorithms produce good convergence with small ensembles of ∼5 members, the parameter convergence is very

slow with medium-sized and large ensembles or the parameters may even diverge. Vice versa, if the algorithms work well with medium-sized and large ensembles, they tend to be unstable with small ensembles.

At least two potential pitfalls are related to bad initialisation of tuning exercises. Firstly, convergence of EPPES suffers from too large initial parameter off-sets, while DE is very robust. For example, in an extreme convergence test, where parameter off-set is an order of magnitude, convergence of EPPES may stop while DE suffers much less. The second pitfall may be

encountered if the initial uncertainty of some parameter is too small respect to the initial off-set. Then convergence to some local optimum is likely. Both algorithms showed that in such a case, the badly initialised parameter remained practically unchanged while the other parameters appeared to compensate the error.

We do not recommend completely blind algorithmic tuning. The parameter off-set should not be excessively large, and the initial parameter off-set and uncertainty should be well proportioned. We also recommend to pay attention to selection of the

tuning algorithm. In case of tuning very uncertain parameters, we recommend to use robust algorithms, which do not suffer from large parameter off-set.

### 4.6  A recipe for successful tuning

Our recipe for economic and efficient tuning is summarised below:

- level 1 of realism (at least initial condition perturbations, possibly also stochastic model perturbations)

- a comprehensive measure is used as a cost function ($\Delta E_m$ in our case)

- a relatively short forecast range is used (24 hours in our case)

- a relatively small ensemble size (20 in our case)

Here we put the recipe into test with more demanding convergence tests. We run four five-parameter tests with TL159, two five-parameter tests with TL399 and one eight-parameter test with TL159 resolution. The parameters in the five-parameter tests

are the same as in the L3 convergence tests. In the eight-parameter test there are three additional parameters from the convection scheme. The parameters are initialised randomly with either 10% too large or too small value, and large uncertainty. The set of initial conditions is the same as before meaning 52 iterations. Here we use EPPES as the optimisation algorithm.

In the six five-parameter convergence tests the parameter values converge toward the default values during the convergence tests in 20 out of 30 cases. In 25 out of 30 cases the final parameter value and the default value are both within the two standard

deviations uncertainty. In the remaining five cases the remaining parameter off-set is slightly more than two standard deviations.





In all 30 cases, the uncertainty of the parameter value decreases during the convergence tests meaning that the parameters do converge even though in some cases they converge to some off-default values.

The parameters tend to converge in a certain way also in these tests. $\theta_2$ tends to converge to a smaller value and $\theta_1$, $\theta_4$ and $\theta_5$ to a larger value than their respective default values.

The results of the eight-parameter convergence test are presented in Figure 7. It shows convergence of the eight parameters in normalised form, and the text boxes in each panel indicate the remaining parameter off-set after 52 iterations. All parameters converge towards their default values. In case of $\theta_5$, the default value is outside the uncertainty range. Additional dimensionality seems to slow down the convergence only a little, which definitely encourages to use algorithmic tuning methods for large parameter sets.

## 5    Discussion

The choice of the cost function is an essential part of the tuning problem. We tested two different cost functions: the root-mean squared error of geopotential at 850 hPa and the moist total energy norm. The former was, as expected, a bad choice, whereas the latter was a better choice. However, the moist total energy norm was not a perfect choice due to the properties of the tunable parameters. The two parameters $\theta_1$ and $\theta_2$ are not equally sensitive to the components of the moist total energy norm; $\theta_1$ is more sensitive to specific humidity and $\theta_2$ to the other components, which may in some cases decrease the overall sensitivity of the moist total energy norm. An option would be to use multiple cost functions, having one dedicated for each tunable parameter, but this leads to other issues instead. Using multiple cost functions would lead to a question of scaling: would each cost function have equal weight or are any of the cost functions considered more important? At the moment we do not have a definitive answer.

In our study, we aimed at finding an optimal set-up for convergence tests by studying different combinations of forecast range and ensemble size. Using an ensemble of 20 members and a forecast range of 24 hours gave the best results. When the ensemble size is too small, the sample size will also be small, which could lead to the case of not having a representative sample. A forecast range of 24 hours seemed optimal. When the forecast range is shorter, $\theta_1$ tends to converge to smaller values and $\theta_2$ to larger values than the default parameter value, whereas a longer forecast leads to $\theta_1$ converging to larger value and $\theta_2$ to smaller value than the default value. The question whether the parameter values depend on the forecast range is profound. The entire forecast range could also be considered, but may lead to similar scaling issues (e.g. Ollinaho et al., 2013) as when using multiple cost functions.

The two optimisers used in this study, EPPES and DE, have different properties. EPPES converges more slowly and estimates the covariance matrix of the parameters, whereas DE gives faster but less steady convergence. The optimisers could therefore be used at different stages of the optimisation: first, DE could be used as a coarse tuner finding the approximate direction, and then EPPES could be used to fine tune the results. This type of tuning process would be of most use when the parameters are known poorly a priori.





We compared the perturbed forecasts against the control forecast run with default parameters. In this case, one would expect that forecasts with default parameters would result in minimum cost function, but this turned out not to be the case. This leads us to the question whether changing the values of the model parameters affect properties of the ensemble, such as its spread. In a well-tuned ensemble prediction system not only should the model be as good as possible (i.e., having optimal parameter values) but the relationship between the spread of the ensemble and the ensemble mean skill should be in balance. We leave this question open for future studies.

## 6    Conclusions

In this paper we have studied the convergence properties of two algorithms used for tuning model physics parameters in a numerical weather prediction model. The tuning process is a computationally demanding task and using an optimal experimental set-up would minimise the amount of computational resources required.

In our experiments we studied two different tuning algorithms and how the convergence properties were affected by (1) the choice of cost function, (2) forecast range, (3) ensemble size, and (4) the complexity of the model set-up (perturbations of initial conditions and stochastic physics turned on or off). In our case, we focused on tuning two parameters of the convection scheme of the OpenIFS model. The model resolution in these tests was T159 (about 125 km).

Our goal was to find an optimal set-up of forecast range and ensemble size with the highest likelihood for fast and reliable convergence; hence, minimising the amount of computations. We ran many convergence tests with different experimental set-ups, calculated the moist total energy norm between the forecasts with perturbed parameters and the control forecast having default parameters values, calculated a fair verification metric (fair-CRPS), and finally compared the experiments against each other. The optimal set-up in our experiments was an ensemble of 20 members and a forecast range of 24 hours.

We tested the optimal set-up for a more complex optimisational task: tuning five and eight parameters at once. In these experiments, the ensemble had 20 members, the forecast range was 24 hours, and the algorithm was run for 52 iterations. Such an experiment would be the same as running a single 1040-day long forecast consuming roughly 400 CPU hours for TL159 model resolution and 4200 CPU hours for TL399 model resolution. These experiments showed that the convergence of most of the parameters was good.

Finally, we conclude our study by answering the question whether algorithmic tuning (of model physics parameters) could be trusted: yes when used with care.

*Code and data availability.*  Basic version of OpenEPS is available under Apache licence version 2.0, January 2004 on Zenodo (https://doi.org/10.5281/zenodo.3759127). Amended version of OpenEPS, which was used in the convergence tests, is also available under Apache licence version 2.0, January 2004 on Zenodo (https://doi.org/10.5281/zenodo.3757601). The amended version contains various modifications such as set-up scripts for EPPES and DE, scripts for calculating cost function values and scripts for processing and plotting output of convergence tests. Besides the archived versions, we encourage to check out also the maintained versions of basic and amended OpenEPS in Github (https://github.com/pirkkao/OpenEPS and https://github.com/laurituppi/OpenEPS). Licence for using OpenIFS NWP



model can be requested from ECMWF user support (openifs-support@ecmwf.int), and the model can be downloaded from ECMWF ftp site (ftp.ecmwf.int). EPPES is available under MIT licence on Zenodo (https://doi.org/10.5281/zenodo.3757580) , and DE is available upon request from vladimir.shemyakin@lut.fi. The initial conditions used in the convergence tests belong to a larger data set. Availability of the data set will be described in (Ollinaho et al., in prep). We want to emphasize that reproducing the results does not require using exactly the same initial conditions than in this paper but any OpenIFS ensemble initial conditions can be used. Output data of the convergence tests is
not archived since it can be easily reproduced.

## 1 Experimental details of EPPES

EPPES needs four hyperparameters: $\boldsymbol{\mu}$, $\boldsymbol{\Sigma}$, $\boldsymbol{W}$, and $n$. The two former describes the initial guess for the distribution of the parameters that are to be estimated, whereas the latter two describes how accurate the initial guess is.

Let $\boldsymbol{\theta} = \{\theta_1, \ldots, \theta_n\}$ be the closure parameters. In EPPES, the prior guess it that the closure parameter follows a Gaussian
distribution $\boldsymbol{\theta_i} \sim \mathcal{N}(\boldsymbol{\mu}, \boldsymbol{\Sigma})$, where $\boldsymbol{\mu}$ is the mean vector of $\boldsymbol{\theta}$ and $\boldsymbol{\Sigma}$ the covariance matrix.

Details of initialisation of the parameter distribution are listed in Table A1 and other settings of EPPES are summarised in Table A2. The mean values are always multiplied with 0.9 or 1.1 in the initialisation of the convergence tests.

## 2 Experimental details of DE

DE requires the boundaries for the parameter search domain to be specified. DE does not explicitly limit any searching di-
rections by default, but some constraints can be specified in order to avoid unfeasible parameter values. In our case, we are targeting to only non-negative values.

Initial search domain is specified in Table A3 and other settings written in the namelist file are summarised in Table A4.

Recalculation step is employed every fifth iteration, it substitutes all usual DE steps (mutation, crossover, selection) and just computes/updates the value of the cost function in the current environment for the elements already in the population.

*Author contributions.* Lauri Tuppi and Pirkka Ollinaho designed the convergence tests, and Lauri Tuppi carried them out. Pirkka Ollinaho provided initial conditions and OpenEPS with comprehensive user support. Vladimir Shemyakin provided hands-on assistance in using DE in the convergence tests. Lauri Tuppi prepared the manuscript with contributions and comments from all co-authors. Especially Heikki Järvinen and Madeleine Ekblom helped with clear formulation of the text. Heikki Järvinen supervised the experimentation and production of the manuscript.

*Competing interests.* The authors declare that they have no conflict of interest.

*Acknowledgements.* The authors are grateful to CSC-IT Center for Science, Finland for providing computational resources, and Juha Lento at CSC-IT for user support with the supercomputers. We thank Olle Räty at Finnish Meteorological Institute for assisting in graphical design



of Figure 3. We are also thankful to Marko Laine at Finnish Meteorological Institute for the insightful discussions about evaluation of convergence tests and user support for EPPES. The figures have been plotted with help of Python Matplotlib library (Hunter, 2007). CDO (Schulzweida, 2019) was used for post-processing the output of OpenIFS.






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

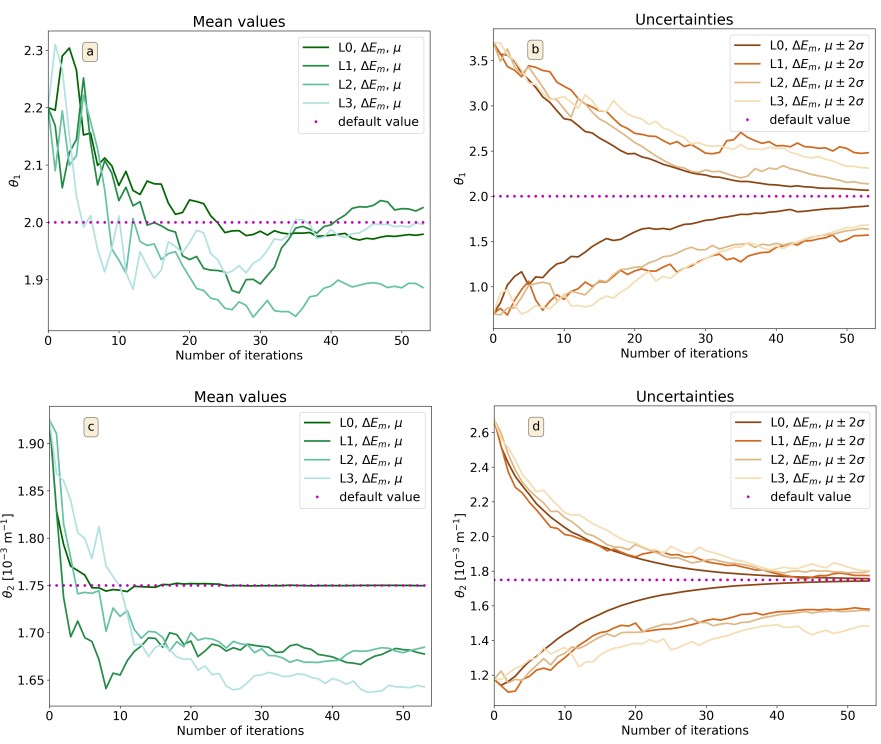

**Figure 1.** Comparison of convergence tests at different levels of realism. Panels (a) and (b) show the evolution of distribution mean value ($\mu$) and the mean value $\pm 2$ standard deviations uncertainty ($\mu \pm 2\sigma$) for $\theta_1$, and (c) and (d) show the same as (a) and (b) but for $\theta_2$. The purple dots show the parameter default values. The x-axes show running number of iterations, i.e., how many ensemble forecasts that have been used. $\Delta E_m$ is used as the cost function, and the levels of realism are summarised in Table 2. EPPES is used as the optimiser, the ensemble size is 50 members and the forecast range 48 hours.





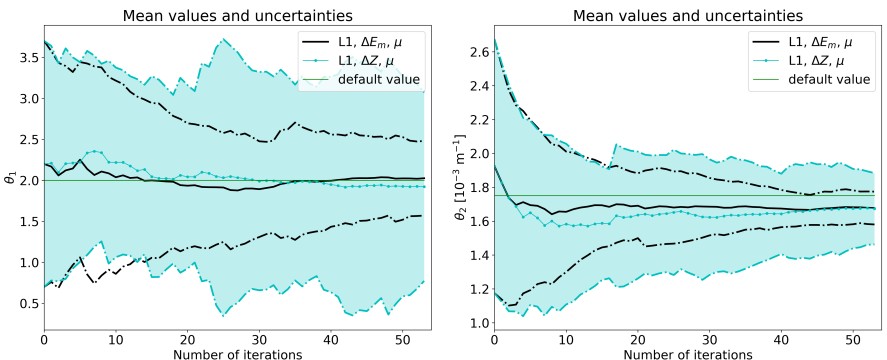

**Figure 2.** Convergence tests with different cost functions. Convergence of $\theta_1$ on the left and $\theta_2$ on the right. The x-axes show running number of iterations. Solid black lines show the evolution of distribution mean values ($\mu$) and black dash-dotted lines the mean values $\pm 2$ standard deviations when $\Delta E_m$ is used as cost function. Cyan dotted lines and shading in the background show the same for $\Delta Z$. Default value shows the fixed parameter value used in the control model. Both convergence tests are L1 tests with 50 ensemble members and 48 hour forecasts. EPPES is used as optimiser.



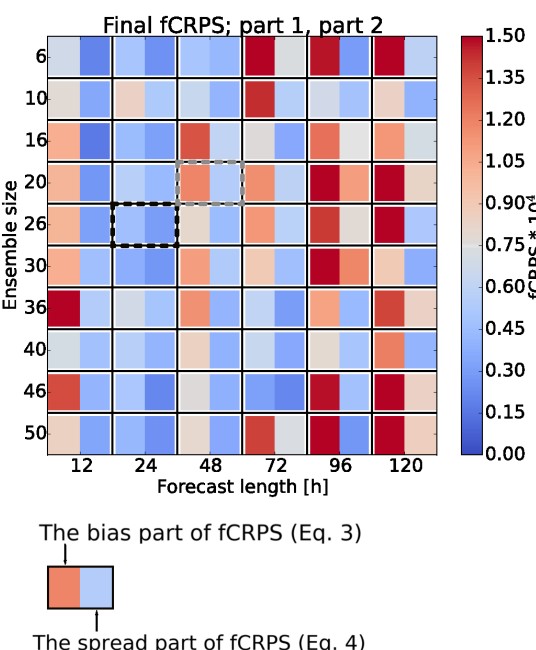

**Figure 3.** Components of fair CRPS from the final iteration of the convergence tests with various forecast ranges and ensemble sizes. In this example, the optimisation algorithm is EPPES and the parameter is $\theta_2$. The left-hand side of each block represents the average distance of the parameter values from the default value (equation 3), and the right-hand side represents the spread of the parameter value distribution (equation 4). Low values and blue colours of both sides of the blocks indicate good convergence.

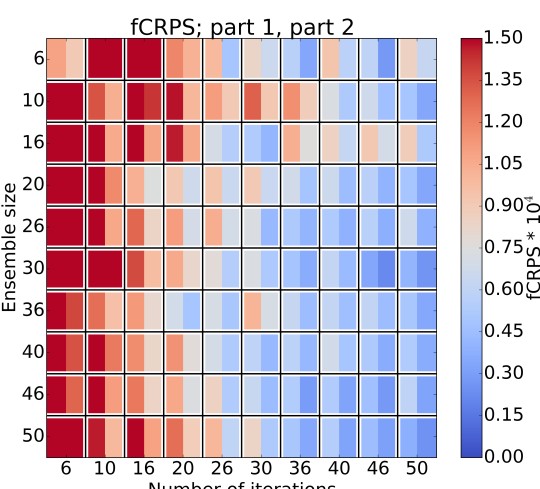

**Figure 4.** Evolution of fCRPS of $\theta_2$ in convergence tests with L2, $\Delta E_m$, EPPES, 24 hour forecasts and various ensemble sizes. The interpretation of the blocks is the same as in Figure 3. The number of iterations indicates how many iterations of the algorithm have been done, or in other words how many ensemble forecasts have been run. Components of fCRPS have been calculated using equations (3) and (4).





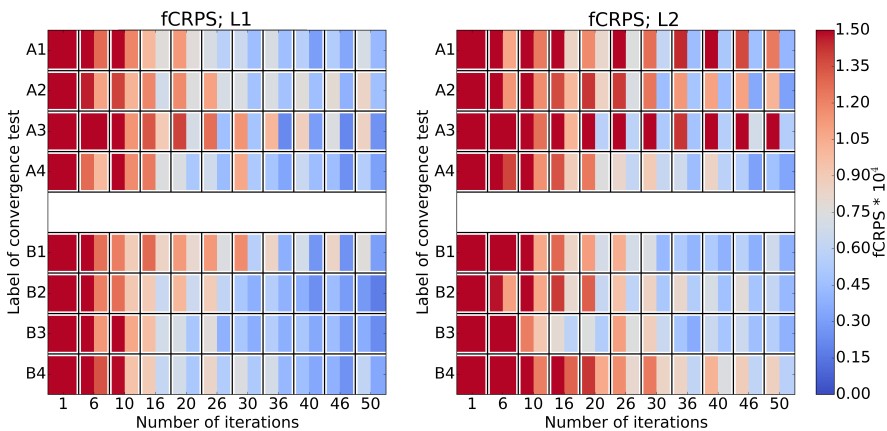

**Figure 5.** Evolution of $\theta_2$ in repeated convergence tests with two selected forecast range – ensemble size combinations highlighted in Figure 3. The level of realism is L1 on the left and L2 on the right. Tests A1 to A4 have been run with forecast range of 48 hours and ensemble size of 20 members, and tests B1 to B4 with 24 hours and 26 members. EPPES was used as an optimiser in these examples. Components of fCRPS have been calculated using equations (3) and (4).





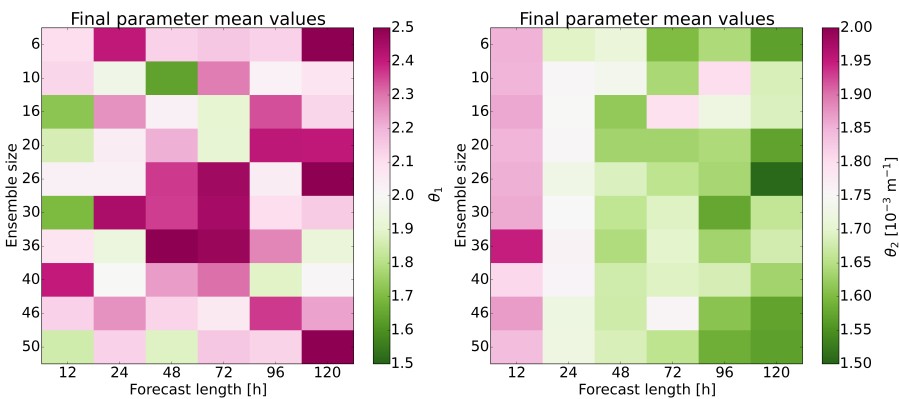

**Figure 6.** Mean values of the parameter distributions proposed by EPPES at the end of the convergence tests. Mean values of $\theta_1$ are on the left and mean values of $\theta_2$ on the right. Purple (green) colour means that the final mean values are larger (smaller) than the default value.



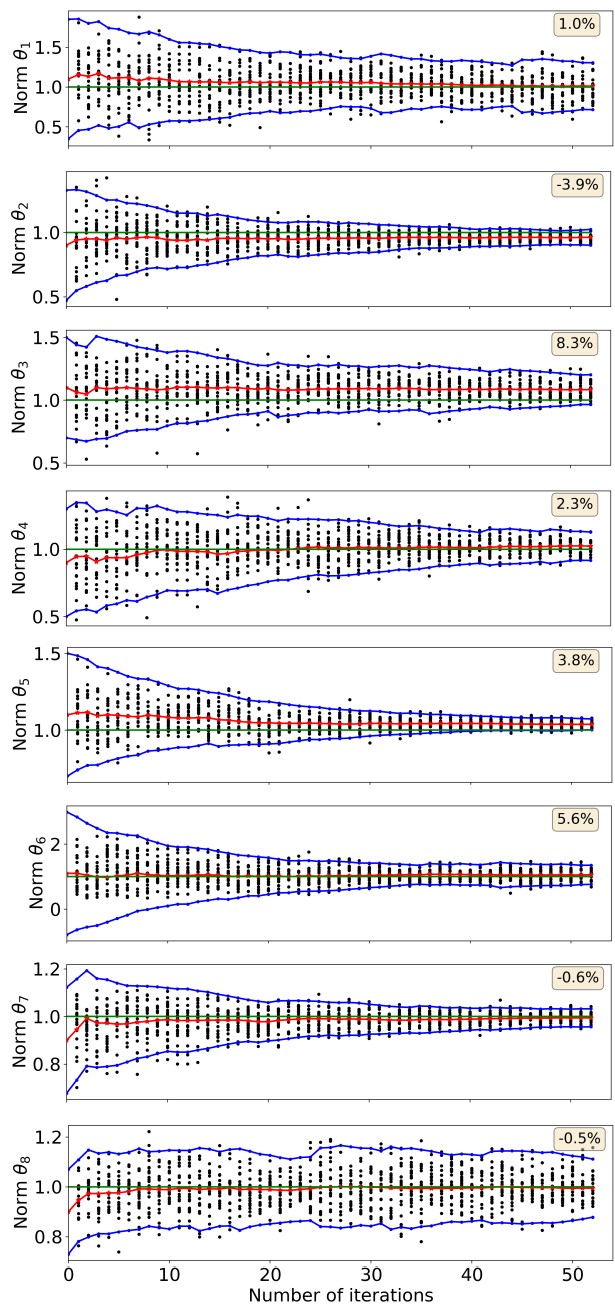

**Figure 7.** Progress of the convergence in the eight-parameter test. The parameter values and uncertainties have been normalised with their default values. Black dots show sampled parameter values, red line with stars shows parameter mean value, blue lines with dots show mean value $\pm 2$ standard deviations and the green line shows the default parameter value that is 1.0 due to the normalisation. The text boxes indicate the remaining parameter off-set, which is the relative distance between the final parameter mean value and the default value. Initial parameter off-set is randomly plus or minus 10 %.





| Parameter | Default value | Short description |
|---|---|---|
| ENTSHALP ($\theta_1$) | 2.0 | Entrainment rate scaling factor for shallow convection |
| ENTRORG ($\theta_2$) | $1.75 \cdot 10^{-3}$ m$^{-1}$ | Entrainment per unit length for deep convection |
| DETRPEN ($\theta_3$) | $0.75 \cdot 10^{-4}$ m$^{-1}$ | Turbulent detrainment per unit length for deep convection |
| RPRCON ($\theta_4$) | $1.4 \cdot 10^{-3}$ s$^{-1}$ | Conversion factor from cloud water/ice to rain/snow |
| RDEPTHS ($\theta_5$) | 20000 Pa | Depth of layer for shallow convection |
| RMFDEPS ($\theta_6$) | 0.3 | Fractional massflux for downdrafts at level of free sinking |
| RHEBC ($\theta_7$) | 0.9 | Critical relative humidity below cloud for evaporation |
| ENTRDD ($\theta_8$) | $3.0 \cdot 10^{-4}$ m$^{-1}$ | Average entrainment per unit length for downdrafts |

**Table 1.** Parameters of the convection scheme of OpenIFS. $\theta_1$ and $\theta_2$ are used the most in this study.





|  | **Number of parameters** | **Different initial conditions** | **Stochastic physics (SPPT)** |
|---|---|---|---|
| **Level 0** (L0) | 2 | No | No |
| **Level 1** (L1) | 2 | Yes | No |
| **Level 2** (L2) | 2 | Yes | Yes |
| **Level 3** (L3) | 5 | Yes | Yes |

**Table 2.** Summary of convergence tests with different degrees of realism.





| Parameter | Mean | Variance | Lower bound | Upper bound |
|---|---|---|---|---|
| ENTSHALP ($\theta_1$) | 2.0 | 0.5625 | 0.5 | 6.0 |
| ENTRORG ($\theta_2$) | 1.75e-3 | 1.40625e-7 | 1e-4 | 1e-2 |
| DETRPEN ($\theta_3$) | 0.75e-4 | 2.25e-10 | 1e-5 | 1e-3 |
| RPRCON ($\theta_4$) | 1.4e-3 | 7.84e-8 | 1e-4 | 1e-2 |
| RDEPTHS ($\theta_5$) | 20000 | 1.6e7 | 1000 | 60000 |
| RMFDEPS ($\theta_6$) | 0.3 | 0.08 | 0.1 | 0.6 |
| RHEBC ($\theta_7$) | 0.9 | 0.01 | 0.5 | 1.0 |
| ENTRDD ($\theta_8$) | 3.0e-4 | 0.65e-9 | 3.0e-5 | 3.0e-3 |

**Table A1.** Initial values of the convection parameters for EPPES.





| Namelist object | Value | Explanation |
|:---:|:---:|:---|
| maxn | 5 | Length of memory in iterations |
| maxstep | 0.05 | Maximum change of parameter mean value in one iteration |
| lognor | 0 | Use log-normal distribution, 0=no |
| useranks | 1 | Ranking of cost function values instead of using values themselves |

**Table A2.** Other settings of EPPES.





| Parameter | Lower bound | Upper bound |
|---|---|---|
| ENTSHALP ($\theta_1$) | 1.0 | 4.0 |
| ENTRORG ($\theta_2$) | 1.25e-3 | 2.25e-3 |
| DETRPEN ($\theta_3$) | 5.0e-5 | 1.0e-4 |
| RPRCON ($\theta_4$) | 1.0e-3 | 1.8e-3 |
| RDEPTHS ($\theta_5$) | 15000 | 25000 |

**Table A3.** Initial parameter value search area of DE.





| Namelist object | Value | Explanation |
|---|---|---|
| F | 0.5 | Control for amplification of differential variation |
| CR | 0.9 | Crossover probability |
| JP | 0.1 | Probability of generation jumping |
| mutation_type | 2 | Use the best parameter vector in mutation |
| scale_factor_type | 5 | Scale factor randomisation scheme |
| F_l | 0.5 | Lower boundary for scale factor F |
| F_u | 1.0 | Upper boundary for scale factor F |
| pop_function | positive | Limits parameters to be positive |
| Jttr | 0.01 | Scale factor randomisation |

**Table A4.** Other settings of DE.