# Peer review of "Necessary conditions for algorithmic tuning of weather prediction models using OpenIFS as an example"

_Geoscientific Model Development, 2020_

## Referee Comment (RC1) · Peter Düben (Referee) · 29 Jul 2020

The paper is providing a recipe how to optimise parameters within an ensemble NWP system. Such an optimisation is very difficult to realise and the paper, which is based on many years of experience in parameter optimisation, provides essential guidelines how to do it properly and should therefore be published. However, the presentation of the paper can still be improved and I provide suggestions in the following. The English language could also be improved.

[Figure]

- l31: "Semi-realistic" What does this actually mean? I guess all tests in the paper are realistic?

- Page 2: I do not find the list of "synopsis guidance" very useful. It is not clear to me what the text in the brackets is meant to be. More detail? The results of this study? Readers should be told somewhere what they should take from this list. I would recommend to replace this list by a list that provides information on the degrees-of-freedoms that are important for optimisation (#parameters, forecast lead time, #ensemble members, minimisation algorithm, #initial conditions, cost function...). You could also think about a table to add information about advantages and disadvantages when increasing or decreasing the degrees-of-freedoms. You could then have another list of the "boundary conditions" including a couple of points from your list (reproducibility required? optimisation target known? computational resources available?...).

- End of introduction: It would help to provide a very brief overview on the sections that will follow.

- Section 2.2.: Maybe I have missed this but do ensemble members use different initial conditions? Ahh, I found it later on in Table 2. But this information seems to be relevant earlier.

- l87-91: I do not understand the discussion. Please re-word and explain jitter and dither... Personally, I would suggest to have more information on the optimisation methods but I leave this decision to the authors.

- l98: Have I missed something? What is the control and what is the perturbed in this case? Do you compare against the default parameter values? Or re-analysis? At l106 you refer to pseudo-observations which seems to be the same as the control? – Ahh, later I understood that you take the default value as

truth and optimise towards it. However, it is still unclear how you define pseudo-observations. Are these point measurements? Or 3D fields? Do you add random noise to represent measurement uncertainty?

- l100: It should be explained why it only probes a "small fraction" of the domain, why it requires interpolation... this may not be clear to all readers.

- l119-120: I do not understand this.

- l124: During the first read, I was not sure whether you are using ensembles where each ensemble member is using a different parameter value. This could be clarified.

- l138: Why do you use different parameters for different tests?

- l188: "discrepancy of model versions" I do not understand this.

- l189: Why should it not converge for a non-linear response if you have enough statistics? At least slowly?

- l268: "In the six five-parameter convergence tests the parameter values converge toward the default values during the convergence tests in 20 out of 30 cases" It took me a while to realise that 5*6=30. This needs more explanation.

- l325: CPU hours or node hours? I guess it would help if you could provide some rough information on the computer that was used.

- Can you add a brief discussion of local or global minima in the parameter optimisation?

- Would the convergence with complexity be an option for optimisation (upgrade from L1→L2→L3→L4 for finer and finer parameter ranges)?

- Is overfitting a problem for lower complexity configurations?

- If you compare against reanalysis or observations, you will often need to use different initial conditions for the different ensemble members to achieve sufficient spread at the beginning of the predictions. How would this influence the discussion around L1-L4?

---

## Referee Comment (RC2) · Joakim Kjellsson (Referee) · 13 Sep 2020

Joakim Kjellsson (Referee)

jkjellsson@geomar.de

Overall:

The manuscript describes and evaluates the usability of algorithmic tuning for weather prediction models, in this case the convection scheme in OpenIFS. The manuscript uses two methods although only one is discussed and tested in detail. I'm curious about the method, in particular when applied to low-resolution versions of atmosphere models, since it could be a good way forward to improve climate models. Given that OpenIFS is used in a few climate models and that similar convection schemes are used in other models, this paper can be of interest also outside the weather forecasting

community.

I recommend this paper for publication after some comments below have been addressed.

Major comments:

The paper is well structured and the results interesting. The writing needs some work, in particular the descriptions of experiments and the latter half of the results section.

I also feel that the paper should clarify right from the start that any closure scheme, in this case convection, is only a parametrisation of the real world, i.e. there is no "true" parameter value.

The choice of 850 hPa geopotential as a cost function seems odd and the authors also state that they expected this to be a bad choice. While its good to compare a bad choice to a good choice, I think it would make more sense to have used something else, e.g. precipitation or total-column water etc. I don't think all the experiments need to be re-done to accommodate this, but I'd like to know exactly why 850 hPa geopotential was chosen and if the authors think some other pressure level or single variable would be more appropriate.

Minor comments:

Overall: The authors use "Firstly" and "Secondly" a few times in the paper. This should be replaced by "First" and "Second".

Line 15: A closure approximates a complex process with an equation and a parameter. As an example, we approximate diffusion/mixing with some high-order derivative and a coefficient. It don't think it's correct to say "the parameter values are not known exactly" because they are only approximations anyway. There is no magic value of e.g. autoconversion rate that will result in a perfect cloud scheme. What you are looking for is the optimal parameter, i.e. the one that results in the smallest error.

Line 45: How about time of year, i.e. is there a seasonal dependence of the optimal parameters. Also, how are the optimal parameters dependent on model resolution, if at all?

Line 54: I'm a bit confused as to the number of forecasts here. Each test is a 51 member ensemble, done for 52 weeks in a year. And Fig 1 looks like you are using 52 iterations. Does this mean 51 members x 52 weeks x 52 iterations > 100 000 forecasts? Or is it 51 members and each weeks is an iteration, i.e. 51 x 52   2600 forecasts? If it is the latter, then your cost function for early iterations are based on how well OpenIFS simulates winter conditions, while iterations   30 are based on how well OpenIFS simulates summer. I could imagine that the optimal parameter and the convergence are quite different in different seasons, so you would probably want to estimate the optimal parameter set for each season separately.

Line 48: Could the authors please add a reference to a paper or documentation for OpenIFS.

Line 54 (contd): I would also really like to see an estimate of the computational cost of these optimisation experiments, e.g. number of core-hours and total number of simulated days.

Line 98: Is "dD = dx * dy"? Then why not use that? Or "dA" for area. Would make more sense.

Line 105: I'm confused about this equation. The individual terms seem to have different units. $u^2$ is m2/s2, while $L^2q^2/(cpT)$ would be in Joule, and the last term I think is in [kg/m3]?

Line 106: In eq (1), primes denote an individual ensemble member. In (2), primes denote the difference between ensemble member and pseudo observations. First, it is confusing that primes denote two different things on the same page. Second, please explain what pseudo-obs are.

Line 120: What is "n"? Is it an index over all tunable parameters?

Line 149: How do the authors measure convergence? Is it done using parameter mean or uncertainty? A "parametric" way could be to smooth curves of mean parameter values and then see where the derivative $\partial\theta/\partial N$ approaches zero, where N is iteration number. Or do you simply judge this by eye here? I agree that L0 seems to converge the fastest (mean stabilises the quickest and uncertainty decreases quickest). On the other hand L1, L2, L3 converge equally fast in terms of the mean parameter value but L1 has the largest uncertainty in Fig 1b.

Line 151 and 160: What is meant by "fully-realistic"? L3 should be the most "realistic" in the sense that it better represents the full complexity of the entire OpenIFS model. "Fully realistic" would be optimising all parameters (probably > 100) in OpenIFS with SPPT?

Line 166: again, "fast convergence" based on what metric?

Line 166: How about changing the sentence to "is explained by two factors. First ... " followed by "Second ...".

Line 171: I would remove this sentence and then start the next sentence with "... recommend using a more comprehensive cost function which accounts for more than one... ".

Line 174: Since you are tuning convection parameters it would be a more fair comparison to use a $\Delta Z$ taken at 200 or 300 hPa. I'm not saying it needs to be done, but it is worth noting that it would be a better comparison.

Line 176: I would change the first sentence to "The process of finding a forecast range ... " and then merge with the next sentence so that it ends "of computational resources is done in two steps." The next sentence would be "First ... forecast ranges. " followed by "Second we take the forecast range ... ensemble size. ".

Line 179: Merge "Using L2 realism" with either the previous or following sentence.

Line 182: I'm struggling to understand what is done here. Do you take the final converged state from Fig 1 c,d, then set parameters to default, and compare how the estimated optimal parameters compare to those you got from Fig 1 c,d? Or do you run a convergence test and compare to the default parameter value as in eq 3? If you are doing the former, please explain this more clearly. If you are doing the latter, then why is low bias good? A large bias could mean that the tuning process worked really well and you got a more optimal parameter value.

Line 186: Could it also be due to the fact that you are focusing on convection parameters and that convection is short lived? If you run 72 hours then you get an influence of synoptic variability which adds extra complexity. If you were tuning orographic wave drag you might want longer forecasts?

Line 194: Change to "We now focus on the forecast range of 24 hours and perform convergence tests with ... ".

Line 195: Change to "Fig 4 indicates that convergence tests with ensemble size > 20 are stable since convergence tests with smaller ensemble sizes do not show... ".

Line 197: What is meant by the two sentences strarting with "sampling variance". What is "sampling variance" here?

Line 198: Change to "Fig 4 also enables comparison ... "

Line 201: Change to "not seem to be necessary to achieve good convergence. "

Line 202: Change to "Results here are for $\theta_2$ but the same conclusions can be drawn from $\theta_1$ ... "

Line 211: I would remove this sentence.

Line 213: The black box In Fig 3 is clearly visible, but the gray one is not. Please use some other colours, e.g. green and yellow, to make sure the boxes stand out. Also explain in the caption for Fig 3 what the boxes are.

Line 217: Rewrite to "shows that both L1 set-up s yield fairly reproducible convergence.".

Line 224: L1 does not use SPPT, i.e not stochastics in OpenIFS. So why are A1-A4 not identical? Are the initial conditons perturbed differently each time, or is there some other stochastic component activated?

Line 226: The language switches from present to past tense here. Please commit to one throughout the paper.

Line 229: I would rewrite sentence to "The opposite is true for $\theta_1$." Also, I would like to see the plots of this (like Fig 1) in a supplementary materials or appendix.

Line 231: I would replace "(to the left)" by Fig 6a and similarly for the other panel. Then I would add letters a,b to the two subplots.

Line 231  232: Words "especially" and "actually" are superfluous and should be removed.

Line 232: It is interesting that $\theta_2$ depends so strongly on forecast length, but it is also interesting how it is not so sensitive to ensemble size. Could this be an effect of not using SPPT so that the system lacks spread? Also, why is $\theta_1$ results all over the place? Could the authors comment on this?

Line 232: I think the rest of this paragraph needs a good rewrite. Here I would rephrase to something like "We now examine the cost functions for two sets of ensemble forecasts (please specify length), one using the default parameters and one using parameters obtained from the optimisation (Fig 6)."

Line 236: Rewrite to "Results show that globally optimal parameter values are different from the their respective default values ... "

Line 241: The sentence includes a statement "we are unsure whether .. " but ends with a question "does the dependency... ". Split the sentences.

Line 241: This is a complete speculation from me: Could it be that you have found parameters that optimise the inter-ensemble spread, i.e. the atmospheric states in the ensemble with optimised parameters might be different from the control forecast, but have small spread which could give a low cost function. For instance if the optimisation produces less convection...

Line 249: If you use "First" in a previous sentence, you should use "Second" here.

Line 250: "with respect to"

Line 250: Merge sentences.

Lines 253-256: This part is very vague. For instance, do you mean that EPPES and DE are both robust? Or is one robust while the other is not? What is excessively large? How can a user know this in advance?

Line 259: This makes sense if I'm only looking to optimise 1-2 variables. But if a forecasting centre wants to optimise their entire convection scheme ( 10 parameters) they would need to go for something larger.

Line 265: How are the five-parameter tests different from the L3 tests before?

Line 266: Which additional three parameters are used?

Line 267: Remove last sentence and make first sentence on line 263 be " ...demanding convergence tests with EPPES."

Lines 269-270: When you say 20 of 30 or 25 of 30, which ones do you mean? Do some variables converge to off-default variables more often than others? A plot or some more text would help here.

Line 270: "uncertainty" = "model spread" ?

Line 270: Do you compute cost function of difference to control run? So we would expect variables to converge to default values?

Line 266: How is the uncertainty chosen? Is this taken from previous optimisation runs, or do you use a guess?

Line 273: This sentence adds no information. Remove it.

Line 279: I disagree that all parameters converge. Parameter 8 does not seem to converge at all. Parameter 1 converges to some extent, but very slowly. How is convergence gauged? By eye? And by what metric, convergence of value or model spread?

Line 282: If it was expected to be bad, why not try another level? 300 hPa would have made more sense maybe, or using surface field like precipitation or total-column precipitable water etc.

Line 284: Could the authors please expand on this? Why are they not equally sensitive?

Line 295: Could it be that some parameters (e.g. convection which is fast) should be tuned with short forecasts while other parameters (e.g. gravity wave drag which is slower) should use longer forecast times?

Line 345: How do you guess the covariance matrix? If you start with a bad guess of the covariance, does that affect the results?

Table 1: It would also be good with a short description of how each parameter should impact the two different cost functions, which would help the argument on line 284.

---

## Author Comment (AC1) · 10 Oct 2020

Necessary conditions for algorithmic tuning of weather prediction models using OpenIFS as an example

Lauri Tuppi, Pirkka Ollinaho, Madeleine Ekblom, Vladimir Shemyakin and Heikki Järvinen

We would like to thank the reviewers for their critical and constructive comments. We agree with most of the comments and we have made changes accordingly.

Structure of the final response is as follows: (1) original comment from referee (2)

[Figure]

author's response (3) author's changes in manuscript.

Comments by Peter Düben are considered first and comments by Joakim Kjellsson after that.

###################################################

Comments by Peter Düben and the authors' answers:

(1) The paper is providing a recipe how to optimise parameters within an ensemble NWP system. Such an optimisation is very difficult to realise and the paper, which is based on many years of experience in parameter optimisation, provides essential guidelines how to do it properly and should therefore be published. However, the presentation of the paper can still be improved and I provide suggestions in the following. The English language could also be improved. (2) We thank for the feedback, and we have made numerous modifications according your comments. The comments of the second reviewer in particular helped to improve the English language. We also found some clear errors, and now we hope that readability of the text has improved. (3) Exact locations of modifications to the manuscript can be found in the comments below. The English language has been improved at least on lines 31, 86, 198, 229, 235, 236, 294-304, 312 and 321.

*[1] (1) l31: "Semi-realistic" What does this actually mean? I guess all tests in the paper are realistic? (2) Different degrees of realism can be thought as a scale. At one end there are idealistic experiments, such as tuning experiments with Lorenz-95 model. At the other end there are fully-realistic tuning experiments with full NWP model and with analysis or other real world data as the reference. Our experiments are semi-realistic since they do not use reference data representing the real world but artificial data generated with a control model with fixed parameter values. Convergence tests can be thought being somewhere between idealistic and fully-realistic experiments. (3) No modifications made to the manuscript.

*[2] (1) Page 2: I do not find the list of "synopsis guidance" very useful. It is not clear to me what the text in the brackets is meant to be. More detail? The results of this study? Readers should be told somewhere what they should take from this list. I would recommend to replace this list by a list that provides information on the degrees-of-freedoms that are important for optimisation (#parameters, forecast lead time, #ensemble members, minimisation algorithm, #initial conditions, cost function...). You could also think about a table to add information about advantages and disadvantages when increasing or decreasing the degrees-of-freedoms. You could then have another list of the "boundary conditions" including a couple of points from your list (reproducibility required? optimisation target known? computational resources available?...). (2) We are sorry to hear this comment because we thought it would be instructive for reader to first get first a synopsis, before dwelling into more details and thus get a feel whether the article answers his/hers concerns. We have made an effort to amend the text to include the Reviewer's message. (3) Lines 32-49 have been rewritten.

*[3] (1) End of introduction: It would help to provide a very brief overview on the sections that will follow. (2, 3) We have added a short introduction for the coming sections. It can be found on lines 50-52.

*[4] (1) Section 2.2.: Maybe I have missed this but do ensemble members use different initial conditions? Ahh, I found it later on in Table 2. But this information seems to be relevant earlier. (2) The purpose of section 2.2. is to present the ensemble forecasting tool that was used in the convergence tests. We agree that readers might want know already at this point that how are the convergence tests run. Section 3 is dedicated to explain the anatomy of the convergence tests and ensemble forecasts therein. We think that this separation is after all the clearest way to first present the tools and then what was done with them. This way the readers will have a fresh image of the structure of the convergence tests in the memory when they enter to the results section. (3) We decided not to modify the manuscript.

*[5] (1) l87-91: I do not understand the discussion. Please re-word and explain jitter

and dither... Personally, I would suggest to have more information on the optimisation methods but I leave this decision to the authors. (2) We have rewritten the details of DE in a different way so that readers not familiar to DE would understand. Jitter and dither act to increase diversity of the parameter population so that they cause small random variation in the parameter values so that DE does not stick to some specific values. Jitter corresponds to the approach when the scale factor is randomized for every mutant parameter vector in the mutation process by sampling from the given (usually uniform) distribution. Dither corresponds to the approach when the fixed scale factor is slightly randomized for each single component of the mutant vector in the mutation process. The two optimisation methods themselves are not in the centre of this study, and very detailed description of the methods can be found from the references. It was mentioned in the introduction that the general guideline of algorithmic tuning should apply to any ensemble based optimisation methods. EPPES and DE are merely examples. We could have used different methods but we had access to these two. (3) Respective modifications can be found on lines 96-103.

*[6] (1) l98: Have I missed something? What is the control and what is the perturbed in this case? Do you compare against the default parameter values? Or re-analysis? At l106 you refer to pseudo-observations which seems to be the same as the control? – Ahh, later I understood that you take the default value as truth and optimise towards it. However, it is still unclear how you define pseudo-observations. Are these point measurements? Or 3D fields? Do you add random noise to represent measurement uncertainty? (2) During the entire paper the comparison is done always against the model with default parameter values. We acknowledge that usage of terms 'control model' and 'pseudo-observations' might be confusing. Pseudo-observations mean output of the control model. Pseudo-observations mean values taken from the control model, and they are 3D fields. For simplicity we do not simulate measurement uncertainty. We unified the terminology. (3) Respective changes can be found on lines 109-111.

\*[7] (1) l100: It should be explained why it only probes a "small fraction" of the domain, why it requires interpolation... this may not be clear to all readers. (2) As $\Delta Z$ probes only 850 hPa level is not sensitive to perturbations of geopotential in middle and upper troposphere where perturbations of deep convection has the most influential effects. Interpolation is required because of the grid structure of OpenIFS. OpenIFS has a terrain following hybrid sigma vertical coordinate, which means that the model levels are not aligned with pressure levels in the lower troposphere. In mountainous areas 850 hPa level is ambiguous since it is below the ground. (3) We have modified the text to explain these points better on lines 115-118.

\*[8] (1) l119-120: I do not understand this. (2) fCRPS consists of two parts; the first part measures how far the ensemble mean value is from the target value and the second part measures how uncertain the ensemble is i.e. how much there is spread. In the original version of fCRPS, the second part is subtracted from the first part. The philosophy of the original fCRPS is that it penalises ensembles having mean value far from the target and small spread, and mean value close to the target and large spread. However, when the mean is far from the target and the spread is large, the original fCRPS gives a good score even though from the tuning point-of-view the result is in fact bad so in some cases the score is misleading. Therefore, we use the parts separately in order to avoid the cases of misleading results. (3) No modifications to the manuscript were made.

\*[9] (1) l124: During the first read, I was not sure whether you are using ensembles where each ensemble member is using a different parameter value. This could be clarified. (2) Each ensemble member uses different parameter value. (3) We have added a clarification to line 142.

\*[10] (1) l138: Why do you use different parameters for different tests? (2) L0, L1 and L2 tests use two parameters ($\theta 1$, $\theta 2$), whereas L3 tests use five parameters ($\theta 1$ to $\theta 5$). There should have been L0 to L2 instead of just L2. (3) The error has been fixed on line 157.

*[11] (1) l188: "discrepancy of model versions" I do not understand this. (2) The initial conditions have been generated with IFS cycle 43r3 while we used OpenIFS, which was part of IFS cycle 40r1. There are substantial differences between the physical parametrisations between the model versions so this "discrepancy of model versions" may have caused stronger-than-usual spin-up at the beginning of the forecasts. ECMWF collects and publishes the model changes in here https://www.ecmwf.int/en/forecasts/documentation-and-support/changes-ecmwf-model (3) We added a reference to the web page on line 62.

*[12] (1) l189: Why should it not converge for a non-linear response if you have enough statistics? At least slowly? (2) We do not argue that it would not eventually converge but we were surprised that there was visible convergence even with this small statistics. Our cost function was of quadratic form so we were expecting that the non-linear nature of the model destroys the signal coming from the parameter perturbations totally in longer forecast ranges but obviously the model develops large enough bias, which is detectable even though there are non-linear perturbations. (3) No modifications were made to the text.

*[13] (1) l268: "In the six five-parameter convergence tests the parameter values converge toward the default values during the convergence tests in 20 out of 30 cases" It took me a while to realise that 5*6=30. This needs more explanation. (2) We have re-worded this and emphasize that all six convergence tests with five parameters are considered at once. (3) We have re-worded the discussion about the six five-parameter tests on lines 292-300.

*[14] (1) l325: CPU hours or node hours? I guess it would help if you could provide some rough information on the computer that was used. (2) It is talking about core hours meaning the wall clock time of execution multiplied by the number of cores so we have re-worded it clearer. The computer used consisted of Intel Haswell nodes having 24 CPUs each. (3) We changed CPUs to cores and added very short information about the computer on lines 361-362.

*[15] (1) Can you add a brief discussion of local or global minima in the parameter optimisation? (2) We added a short discussion. Local minimisation of the cost function may happen in algorithmic tuning mainly if the initialisation of the parameter values is bad. If some parameter has too small uncertainty and too large distance to the optimum, other parameters may compensate the error, and the cost function becomes locally minimised. The issue about converging to different values depending on the forecast range is a different point. When the parameters are initialised appropriately and initial condition perturbations active, sticking to some local minimum of the cost function is unlikely. (3) We have added a short discussion on lines 334-339.

*[16] (1) Would the convergence with complexity be an option for optimisation (upgrade from L1→L2→L3→L4 for finer and finer parameter ranges)? (2) This is one option, which can be used. Especially if the number of parameters is relatively large (say 20), this could be very efficient way to make a shortcut into fine tuning of the parameters. However, the uncertainty of the parameters, which remains at the end, increases as complexity increases meaning that one must be careful to not pose too small parameter ranges because inflation of parameter uncertainty often proves to be more difficult than deflation at least for EPPES. Therefore, the user could get too small uncertainty at the end. (3) No changes were made regarding this comment.

*[18] (1) Is overfitting a problem for lower complexity configurations? (2) We agree that overfitting is very likely a problem, and this is actually one reason why we do not recommend to use the configuration with the lowest complexity (L0) as such in fully-realistic tuning. 'Overfitting' is better wording than ambiguous 'unwanted results' so we re-worded some places a bit. (3) We have changed 'unwanted results' to 'overfitting of the parameters' on line 174.

*[19] (1) If you compare against reanalysis or observations, you will often need to use different initial conditions for the different ensemble members to achieve sufficient spread at the beginning of the predictions. How would this influence the discussion around L1-L4? (2) Only those convergence tests with the lowest level of complexity

(L0) use the same initial conditions. We mention in the text that L0 should be used only to check that the optimisation infrastructure does not have major flaws. In all other convergence tests (L1, L2 and L3) the initial condition perturbations are active so every ensemble member already has unique initial conditions. We also acknowledge that ensemble initial conditions sample the weather states better, and this was the reason to use them in L1, L2 and L3. (3) No changes were made regarding this comment.

############################################################

Comments by Joakim Kjellson and authors' responses:

(1) Overall: The manuscript describes and evaluates the usability of algorithmic tuning for weather prediction models, in this case the convection scheme in OpenIFS. The manuscript uses two methods although only one is discussed and tested in detail. I'm curious about the method, in particular when applied to low-resolution versions of atmosphere models, since it could be a good way forward to improve climate models. Given that OpenIFS is used in a few climate models and that similar convection schemes are used in other models, this paper can be of interest also outside the weather forecasting community. I recommend this paper for publication after some comments below have been addressed. (2) We thank for the insightful comments. In particular, we thank for the effort in making concrete suggestions how to re-word certain sections of the manuscript better. We are glad to hear that our paper is potentially interesting also for climate modellers. (3) Exact locations of modifications to the manuscript can be found in the comments below.

*[1] (1) The paper is well structured and the results interesting. The writing needs some work, in particular the descriptions of experiments and the latter half of the results section. (2) We have worked on making the unclear sections more readable. Several small and a couple of larger modifications have been made according to the comments below. We now hope that the manuscript is now easier to understand. (3) Exact locations are shown in the comments below.

*[2] (1) I also feel that the paper should clarify right from the start that any closure scheme, in this case convection, is only a parametrisation of the real world, i.e. there is no "true" parameter value. (2) This is a very good point. The difference between the real world convection and parametrisation might not be clear to all readers so stating this point shortly is useful. (3) Modifications can be found on lines 14-16.

*[3] (1) The choice of 850 hPa geopotential as a cost function seems odd and the authors also state that they expected this to be a bad choice. While its good to compare a bad choice to a good choice, I think it would make more sense to have used something else, e.g. precipitation or total-column water etc. I don't think all the experiments need to be re-done to accommodate this, but I'd like to know exactly why 850 hPa geopotential was chosen and if the authors think some other pressure level or single variable would be more appropriate. (2) We acknowledge that using a different level or different variables would have likely made the "bad" cost function better. The motivation for using this not-so-suitable cost function is two-fold: 1) We want to illustrate that simply targeting a single skill score of the system (ability to do give skillful predictions targeting a single model output field) is not enough for optimization purposes as it leaves too much room for compensating errors to take place. This is also highlighted in earlier work with EPPES in IFS and ECHAM5 contexts (Ollinaho et al. 2013a, b). We believe that had the choice been precipitation or TCWV instead of 850hPa pressure it would not really have affected the results. We motivate this line of thinking from the procedures of operational weather prediction centres, where model forecast skill changes are verified from multiple perspectives with a holistic view during model code updates (i.e. using different skill scores and multiple model fields). 2) We want to highlight that choosing the target for the optimization carefully really matters, in this sense having a very radical contrast helps to deliver the message. We also do show how fine tuning of the target for optimization also makes a difference (the forecast length at which we target the moist energy norm to be minimized). (3) We have emphasized that it was an intentional choice to make a radical comparison on lines 308-309.

*[4] (1) Overall: The authors use "Firstly" and "Secondly" a few times in the paper. This should be replaced by "First" and "Second". (2) We have changed all of these words accordingly. (3) Modifications can be found on lines 113, 116, 188, 189, 199, 251 and 271.

*[5] (1) Line 15: A closure approximates a complex process with an equation and a parameter. As an example, we approximate diffusion/mixing with some high-order derivative and a coefficient. I don't think it's correct to say "the parameter values are not known exactly" because they are only approximations anyway. There is no magic value of e.g. autoconversion rate that will result in a perfect cloud scheme. What you are looking for is the optimal parameter, i.e. the one that results in the smallest error. (2) We totally agree with the reviewer's comment: what we search is the "optimal" parameter value rather than some hypothetical "correct" value. The text has been corrected to accommodate this comment. (3) Modifications can be found on lines 14-16.

*[6] (1) Line 45: How about time of year, i.e. is there a seasonal dependence of the optimal parameters. Also, how are the optimal parameters dependent on model resolution, if at all? (2) We have been thinking about the seasonality of the parameters but we ended up in a conclusion that if any seasonality existed, it would mean that there would be some fundamental deficiency in the parametrisation itself. In case of OpenIFS we assume that the model developers have accounted for this possibility so we do not expect any seasonality in the parameter values. The effect of model resolution is a different case. Mauritsen et al. (2012) state that the model parameters must be re-tuned after every modification of the model. Upgrade of resolution is a major modification. However, it depends case-by-case how model resolution affects each model parameter but changes of a few percentage units are likely. Moreover, in our experiments, we do not expect seasonality since the reference is a control model with fixed known parameter values. (3) No modifications made regarding this comment.

*[7] (1) Line 54: I'm a bit confused as to the number of forecasts here. Each test is

a 51 member ensemble, done for 52 weeks in a year. And Fig 1 looks like you are using 52 iterations. Does this mean 51 members x 52 weeks x 52 iterations > 100 000 forecasts? Or is it 51 members and each weeks is an iteration, i.e. 51 x 52 2600 forecasts? If it is the latter, then your cost function for early iterations are based on how well OpenIFS simulates winter conditions, while iterations 30 are based on how well OpenIFS simulates summer. I could imagine that the optimal parameter and the convergence are quite different in different seasons, so you would probably want to estimate the optimal parameter set for each season separately. (2) In Fig 1 there are 50+1 member ensemble run once a week throughout the year meaning 52 ensemble forecasts. Therefore, the total number of forecasts is 51 x 52 = 2600 forecasts. We have two reasons to not expect the parameter values to depend on the season. The first one is that we expect that the developers of OpenIFS have taken any possible seasonalities into account in designing the parametrisations. The second one is that we use control model with fixed parameter values to generate the reference data used in the convergence tests. We do not use analyses or reanalyses in this study. (3) No modifications made regarding this comment.

*[8] (1) Line 48: Could the authors please add a reference to a paper or documentation for OpenIFS. (2) Standalone documentation of OpenIFS does not exist. However, the documentation of IFS can be used instead. (3) Reference to documantation of IFS cycle 40r1 added on line 56.

*[9] (1) Line 54 (contd): I would also really like to see an estimate of the computational cost of these optimisation experiments, e.g. number of core-hours and total number of simulated days. (2) There already is a rough estimate of core hours and number of simulated days used in two experiments in lines (3) No modifications made regarding this comment.

*[10] (1) Line 98: Is "dD = dx * dy"? Then why not use that? Or "dA" for area. Would make more sense. (2) We decided to use uniform notation for equations [1] and [2]. Notation "dD" is used in equation [2.2] of Ehrendorfer et al. (1999), and we decided to

use the same notation. (3) No modifications made regarding this comment.

*[11] (1) Line 105: I'm confused about this equation. The individual terms seem to have different units. u 2 is m2/s2, while L 2 q 2 /(cpT ) would be in Joule, and the last term I think is in [kg/m3]? (2) We acknowledge that the moist total energy norm is a collection of different cost functions scaled so that their weight would be comparable. This also leads to different units with different terms. As the moist total energy norm is somewhat artificial construction (Marquet et al. 2020), which does not measure energy as such, units are not important. Our formulation of moist total energy norm follows equation [2.2] of Ehrendorfer et al. (1999) with the exception that the surface pressure is treated as in Ollinaho et al. (2014). This comment made us to notice that the temperature term (cp/Tr)*T'2 was missing from the equation, and we added it. Moreover, we have divided the resulting number with the mass of the atmosphere so instead of 1/2 there should be 1/(2Ma) where Ma is the mass of the atmosphere. Our calculations have been performed with the temperature term and division by the mass of the atmosphere. Only the formula was wrong. (3) We added the missing terms and explanations on lines 121 and 124.

*[12] (1) Line 106: In eq [1], primes denote an individual ensemble member. In [2], primes denote the difference between ensemble member and pseudo observations. First, it is confusing that primes denote two different things on the same page. Second, please explain what pseudo-obs are. (2) Yes, the notation is confusing. We have replace prime with asterisk in equation [1]. Pseudo-observations mean the reference data generated with the fixed control model. Pseudo-observations are 3D fields. We have added a more clear explanation for the pseudo-observations. (3) Modifications can be found on lines 109-112.

*[13] (1) Line 120: What is "n"? Is it an index over all tunable parameters? (2) Yes, "n" is an index over all tunable parameters. (3) We added explanation on line 141.

*[14] (1) Line 149: How do the authors measure convergence? Is it done using pa-

rameter mean or uncertainty? A "parametric" way could be to smooth curves of mean parameter values and then see where the derivative $\partial\theta/\partial N$ approaches zero, where N is iteration number. Or do you simply judge this by eye here? I agree that L0 seems to converge the fastest (mean stabilises the quickest and uncertainty decreases quickest). On the other hand L1, L2, L3 converge equally fast in terms of the mean parameter value but L1 has the largest uncertainty in Fig 1b. (2) Here the convergence is judged by eye from Figure 1. We judge the convergence in two ways: first, how far are the mean values of both parameters from the default values, and second, how large are the uncertainties at the end of the convergence tests. The target is that the parameter mean value becomes the same as the default value, and uncertainty becomes smaller than at the beginning. Therefore, we think that parametric way to measure convergence is not needed. It is true that in L1 test parameter 1 happens to retain large uncertainty during that test. However, we expect that this is caused by random variation during the test as each test has been done only once in Figure 1. Parameter 2 shows less signs of random variation as it is more sensitive to the cost function than parameter 1. (3) No modifications made regarding this comment.

*[15] (1) Line 151 and 160: What is meant by "fully-realistic"? L3 should be the most "realistic" in the sense that it better represents the full complexity of the entire OpenIFS model. "Fully realistic" would be optimising all parameters (probably > 100) in OpenIFS with SPPT? (2) Here "fully-realistic" means using analyses, reanalyses or observations as the reference data. It is true that the usage of word "fully-realistic" might be confusing. We have checked all sentences where the word has been used, and reworded if it was used for something else than referring to "genuine" model tuning with real observations or analyses as the reference data. We changed the word "realism" to "complexity" in order to decrease the likelihood of confusion. (3) Modifications can be found on lines 148, 149, 156, 168-170, 173, 177, 183, 200, 201, 243, 283, 351, caption of Figure 1 and 5, and Table 1 .

*[16] (1) Line 166: again, "fast convergence" based on what metric? (2) The convergence has been judged by eye from Figure 2: how far are the final parameter values from the default values, and how much uncertainty is left. (3) No modifications made regarding this comment.

*[17] (1) Line 166: How about changing the sentence to "is explained by two factors. First ... " followed by "Second ...". (2, 3) We have modified accordingly lines 188 and 189.

*[18] (1) Line 171: I would remove this sentence and then start the next sentence with "... recommend using a more comprehensive cost function which accounts for more than one... ". (2, 3) We have modified accordingly line 193.

*[19] (1) Line 174: Since you are tuning convection parameters it would be a more fair comparison to use a $\Delta Z$ taken at 200 or 300 hPa. I'm not saying it needs to be done, but it is worth noting that it would be a better comparison. (2) The point of choosing $\Delta Z$ with 850 hPa level was to show an example of a simple but very bad cost function. This was an intentional selection for the comparison against the moist total energy norm. We acknowledge that 200 or 300 hPa could lead to a better cost function but it would not change the message we want to deliver. The point is to show that putting effort on careful selection of cost function pays off. Comparison pros and cons of multiple different cost functions is out of scope of this paper. (3) No additional modifications were made.

*[20] (1) Line 176: I would change the first sentence to "The process of finding a forecast range ... " and then merge with the next sentence so that it ends "of computational resources is done in two steps." The next sentence would be "First ... forecast ranges. " followed by "Second we take the forecast range ... ensemble size. ". (2, 3) We have modified lines 199-200 as suggested.

*[21] (1) Line 179: Merge "Using L2 realism" with either the previous or following sentence. (2) We have merged it to the previous sentence. (3) Modifications can be found on line 200.

*[22] (1) Line 182: I'm struggling to understand what is done here. Do you take the final converged state from Fig 1 c,d, then set parameters to default, and compare how the estimated optimal parameters compare to those you got from Fig 1 c,d? Or do you run a convergence test and compare to the default parameter value as in eq 3? If you are doing the former, please explain this more clearly. If you are doing the latter, then why is low bias good? A large bias could mean that the tuning process worked really well and you got a more optimal parameter value. (2) Here we take the final parameter values proposed by EPPES and then use equations [3] and [4] to calculate how much the mean value of the parameter deviates from the default value, and how much there is spread among the parameter values so we are doing the latter. Bias is an absolute value of the distance between the mean and default values. Spread is a scaled absolute value of how far each parameter value is from the other values within the ensemble. In our convergence tests bias is always calculated respect to the default parameter value, which is the target of convergence. Therefore, low bias means that the average of the proposed parameter values is close to the target, which means that location of the parameter value distribution is good. (3) No modifications were made to the text.

*[23] (1) Line 186: Could it also be due to the fact that you are focusing on convection parameters and that convection is short lived? If you run 72 hours then you get an influence of synoptic variability which adds extra complexity. If you were tuning orographic wave drag you might want longer forecasts? (2) Our explanation and your comment actually mean the same. Convection is short lived, which means that the response to perturbations stays approximately linear for a relatively short time. We have not studied the behaviour of other parameters besides the convection parameters but we think that it is likely that different forecast range would be optimal for different parametrisation schemes. (3) No modifications were made to the text.

*[24] (1) Line 194: Change to "We now focus on the forecast range of 24 hours and perform convergence tests with ... ". (2, 3) We have re-worded line 219 accordingly.

*[25] (1) Line 195: Change to "Fig 4 indicates that convergence tests with ensemble size > 20 are stable since convergence tests with smaller ensemble sizes do not show... ". (2, 3) We have re-worded lines 220-221 but changed "since" to "while".

*[26] (1) Line 197: What is meant by the two sentences starting with "sampling variance". What is "sampling variance" here? (2) Here "sampling variance" refers to the random variability caused by small sample size: if the ensemble size is small, on some iterations all truly good parameter values might lead to bad scores by chance, which hampers the convergence of the parameters. (3) No modifications were made to the text.

*[27] (1) Line 198: Change to "Fig 4 also enables comparison ... " (2, 3) We have modified line 223 accordingly.

*[28] (1) Line 201: Change to "not seem to be necessary to achieve good convergence. " (2, 3) We have modified line 226 accordingly.

*[29] (1) Line 202: Change to "Results here are for $\theta$ 2 but the same conclusions can be drawn from $\theta$ 1 ... " (2, 3) We have modified line 226-227 accordingly.

*[30] (1) Line 211: I would remove this sentence. (2,3) We removed the sentence from line 238.

*[31] (1) Line 213: The black box In Fig 3 is clearly visible, but the gray one is not. Please use some other colours, e.g. green and yellow, to make sure the boxes stand out. Also explain in the caption for Fig 3 what the boxes are. (2) We have replace the colours with green and yellow. We have also added an explanation that the boxes highlight the tests repeated in Figure 5. (3) Caption of Figure 3 has been modified.

*[32] (1) Line 217: Rewrite to "shows that both L1 set-up s yield fairly reproducible convergence.". (2, 3) We have modified line 242 accordingly.

*[33] (1) Line 224: L1 does not use SPPT, i.e not stochastics in OpenIFS. So why are A1-A4 not identical? Are the initial conditions perturbed differently each time, or is

there some other stochastic component activated? (2) The reason why even L1 or L0 test are not identical is that EPPES takes the random number seed from the wall clock time so the proposed parameter values are different in each convergence test. (3) No modifications were made to the text.

*[34] (1) Line 226: The language switches from present to past tense here. Please commit to one throughout the paper. (2) We changed all those part to present tense where the change was meaningful. However, in some parts of the text, we use the past tense as a rhetoric means or when we extend discussion of experiments done. In Section 1 we use the past tense in order to advertise what we found. Section 4.5. begins with discussion about tests performed in Sections 4.1. to 4.4. so beginning the section with past tense is natural. Most of Sections 5 and 6 also discuss about the tests in sections 4.1.-4.5. so in those parts using past tense is also natural. (3) Modifications were made on lines 252, 275, 276 and 324.

*[35] (1) Line 229: I would rewrite sentence to "The opposite is true for $\theta\_1$ ." Also, I would like to see the plots of this (like Fig 1) in a supplementary materials or appendix. (2) We have modified the sentence. In our opinion additional figures are not needed since the way how the parameters tend to converge respect to forecast range is clearly seen in Figure 6, which shows the end results of the array of L2 convergence tests. Lines 253-257 serve as an introduction to Figure 6, which is discussed in further details after that. (3) The sentence has been rewritten on line 254.

*[36] (1) Line 231: I would replace "(to the left)" by Fig 6a and similarly for the other panel. Then I would add letters a,b to the two subplots. (2, 3) We have modified the figure, the figure caption and lines 255-256 accordingly.

*[37] (1) Line 231 232: Words "especially" and "actually" are superfluous and should be removed. (2,3) We have removed those words from the entire manuscript (lines 258, 259, 267 and 396).

*[38] (1) Line 232: It is interesting that $\theta\_2$ depends so strongly on forecast length, but

it is also interesting how it is not so sensitive to ensemble size. Could this be an effect of not using SPPT so that the system lacks spread? Also, why is $\theta\_1$ results all over the place? Could the authors comment on this? (2) We did some convergence tests both with SPPT switched on and off on to see if the parameters then behave similarly as in Figure 6. SPPT does not have a large effect on how the parameter mean values tend to converge so it is unlikely that spread has a large role in the dependence. The amount of spread in ensemble forecasts used for parameter tuning is not as important as in operational forecasting for example. Spread itself does not help to find good parameter values. In fact, Figure 5 shows that SPPT merely hinders the convergence as the uncertainty decreases slower. The cost function is less sensitive to variation of $\theta\_1$ so there is also more variability in the results. Sensitivity of the parameters is discussed further in Section 5. (3) Lines 312-316 have been modified.

*[39] (1) Line 232: I think the rest of this paragraph needs a good rewrite. Here I would rephrase to something like "We now examine the cost functions for two sets of ensemble forecasts (please specify length), one using the default parameters and one using parameters obtained from the optimisation (Fig 6)." (2) We have rewritten accordingly. The length of the ensemble forecasts is six days, but $\Delta E\_m$ is evaluated every six hours in order to see how it behaves respect to forecast length. (3) We have modified lines 257-258 accordingly.

*[40] (1) Line 236: Rewrite to "Results show that globally optimal parameter values are different from the their respective default values ... (2, 3) We have modified lines 260-261 accordingly.

*[41] (1) Line 241: The sentence includes a statement "we are unsure whether .. " but ends with a question "does the dependency...". Split the sentences. (2, 3) We have modified lines 264-265 accordingly.

*[42] (1) Line 249: If you use "First" in a previous sentence, you should use "Second" here. (2, 3) We have modified lines 271 and 273.

\*[43] (1) Line 250: "with respect to" (2, 3) We have modified line 274 accordingly.

\*[44] (1) Line 250: Merge sentences. (2, 3) We have modified lines 274-275 accordingly.

\*[45] (1) Lines 253-256: This part is very vague. For instance, do you mean that EPPES and DE are both robust? Or is one robust while the other is not? What is excessively large? How can a user know this in advance? (2) - We mean that one needs to know and understand the nature and main points of the parametrisation to be tuned. Even the best tools available will not work without careful planning of the tuning experiments. Both of the algorithms are robust but DE is more robust than EPPES since it can converge parameters that are further away from the optimum than EPPES. - Usually in tuning of NWP models, the parameter off-set is less than 10%. However, it can be larger in a case of a completely new parametrisation scheme. With excessively large off-set, we mean of an order of magnitude off-set. - In case of re-tuning old parametrisation schemes, excessively large initial parameter off-set is very unlikely as the old optimal values are known, and usually the change is less than 10%. At least theoretically in case of new parametrisation schemes, some parameters might be difficult to quantify with other methods than trial-and-error, so then the user can expect large initial off-set. In this case we recommend visual inspection of the model output before tuning; the output must be physically realistic. (3) No modifications were made to the text.

\*[46] (1) Line 259: This makes sense if I'm only looking to optimise 1-2 variables. But if a forecasting centre wants to optimise their entire convection scheme ( 10 parameters) they would need to go for something larger. (2) We want to point out that we designed the recipe with tow parameters but tested it with five and eight parameters. The results with eight parameters are shown in Figure 7, and the convergence is satisfactory. (3) No modifications were made to the text.

\*[47] (1) Line 265: How are the five-parameter tests different from the L3 tests before? (2) The only difference to L3 tests before is that SPPT is deactivated. (3) No modifications were made to the text.

*[48] (1) Line 266: Which additional three parameters are used? (2) They are explained in Table 1 (RMFDEPS, RHEBC and ENTRDD). (3) We added reference to Table 1 on line 290.

*[49] (1) Line 267: Remove last sentence and make first sentence on line 263 be " ...demanding convergence tests with EPPES." (2, 3) We have modified line 287 accordingly.

*[50] (1) Lines 269-270: When you say 20 of 30 or 25 of 30, which ones do you mean? Do some variables converge to off-default variables more often than others? A plot or some more text would help here. (2) Mean values of both $\theta\_1$ and $\theta\_3$ converge to off-default values for four times, and of $\theta\_4$ for two times. The mean value of each parameter is more than two standard deviations away from the default value once in the six tests. We have added a couple of sentences about those parameters converging the worst. (3) Modifications have been made on lines 292-300.

*[51] (1) Line 270: "uncertainty" = "model spread" ? (2) Uncertainty means the width of the parameter distribution proposed by EPPES. (3) No modifications were made to the text.

*[52] (1) Line 270: Do you compute cost function of difference to control run? So we would expect variables to converge to default values? (2) Yes, control model with default parameter values is used as the reference throughout the paper so we would expect the parameters to converge to their default values. (3) No modifications were made to the text.

*[53] (1) Line 266: How is the uncertainty chosen? Is this taken from previous optimisation runs, or do you use a guess? (2) The initial uncertainty is a guess. However, in order to be conservative, we tested that the initial uncertainty is unlikely to allow completely unrealistic model states prior to the convergence tests. (3) No modifications were made to the text.

*[54] (1) Line 273: This sentence adds no information. Remove it. (2) We removed it from here, and also moved the following sentence to the discussion about convergence to off-default values above. (3) The remaining sentence was moved to lines 295-296.

*[55] (1) Line 279: I disagree that all parameters converge. Parameter 8 does not seem to converge at all. Parameter 1 converges to some extent, but very slowly. How is convergence gauged? By eye? And by what metric, convergence of value or model spread? (2) Looking at Figure 7, also parameter 8 converges: at the end the off-set is smaller than at the beginning and also the uncertainty decreases about 29%. However, you are correct that parameters 8 and 1 show least convergence. Convergence is judged visually from convergence of mean values (= whether they come closer to the default values or not) and decreases of spread of the parameter distributions. (3) No modifications were made to the text.

*[56] (1) Line 282: If it was expected to be bad, why not try another level? 300 hPa would have made more sense maybe, or using surface field like precipitation or total-column precipitable water etc. (2) The root-mean squared error of 850 hPa geopotential was intentionally chosen to be an example of a cost function, which is very simple but does not work in this case. With the chosen example, we showed that careful selection of the cost function pays off. We think that choosing different level would not have changed the message. One must pay attention to the selection of suitable cost function. More comprehensive cost functions tend to perform better. (3) Small changes can be found on lines 308-309.

*[57] (1) Line 284: Could the authors please expand on this? Why are they not equally sensitive? (2) In OpenIFS $\theta\_1$ is active only in the 200 hPa layer beginning from the Earth's surface. This means that the direct effect of perturbation of $\theta\_1$ is confined to the lower troposphere. Moreover, the contribution of $\theta\_1$ to the cost function comes

almost solely from the moisture term as shallow entrainment controls how the moisture distributes in the lower troposphere. $\theta\_2$ controls entrainment of deep convection so the effect is felt in entire troposphere. Perturbations of deep convection affect directly wind and temperature besides humidity, and these fields have been included into moist total energy norm. We added a short text to explain the main points. (3) Modifications can be found on lines 312-316.

*[58] (1) Line 295: Could it be that some parameters (e.g. convection which is fast) should be tuned with short forecasts while other parameters (e.g. gravity wave drag which is slower) should use longer forecast times? (2) This might be possible if the user wishes to tune only one parametrisation scheme at once. However, if the user wishes to tune entire model at once, there is one potential obstacle: the fast perturbations of convection become non-linear and also make the signal of gravity wave drag untraceable when longer forecast time is used. How to make an optimal compromise is left for future studies. (3) No modifications were made to the text.

*[59] (1) Line 345: How do you guess the covariance matrix? If you start with a bad guess of the covariance, does that affect the results? (2) In the initialisation, only the diagonal is filled with variances, and all covariances are set to zero. In guessing of the initial variances, we made sure that $\pm2$ standard deviation perturbation of the parameter does to lead to visually unrealistic model state. If the initial guess is very bad, then other parameters try to compensate the error leading to large jumps of parameter values at the beginning. The parameters may also converge to grossly off-default values. (3) No modifications were made to the text.

*[60] (1) Table 1: It would also be good with a short description of how each parameter should impact the two different cost functions, which would help the argument on line 284. (2) In many cases it is not intuitive how the model parameters affect the cost function so figuring out the connections would require a substantial amount of experimentation.We disagree that in-detail studying of the processes how each parameter affects the model fields used by the cost functions falls within the scope of this study.

In general, convection is not directly linked to 850 hPa geopotential but it is connected in a way or another either to wind, temperature, humidity or surface pressure. (3) No modifications were made to the text.

References

Ehrendorfer, M., Errico, R. M., and Raeder, K. D.: Singular-Vector Perturbation Growth in a Primitive Equation Model with Moist Physics, Journal of the Atmospheric Sciences, 56, 1627–1648, https://doi.org/10.1175/1520-0469(1999)056<1627:SVPGIA>2.0.CO;2, https://doi. org/10.1175/1520-0469(1999)056<1627:SVPGIA>2.0.CO;2, 1999.

Marquet, P., Mahfouf, J.-F., and Holdaway, D.: Definition of the Moist-Air Exergy Norm: A Comparison with Existing "Moist Energy Norms", Monthly Weather Review, 148, 907–928, https://doi.org/10.1175/MWR-D-19-0081.1, https://doi.org/10.1175/MWR-D-19-0081.1, 2020.

Mauritsen, T., Stevens, B., Roeckner, E., Crueger, T., Esch, M., Giorgetta, M., Haak, H., Jungclaus, J., Klocke, D., Matei, D., Mikolajewicz, U., Notz, D., Pincus, R., Schmidt, H., and Tomassini, L.: Tuning the climate of a global model, Journal of Advances in Modeling Earth Systems, 4, https://doi.org/10.1029/2012MS000154, https://agupubs.onlinelibrary.wiley.com/doi/abs/10.1029/2012MS000154, 2012.

Ollinaho, P., Bechtold, P., Leutbecher, M., Laine, M., Solonen, A., Haario, H., and Järvinen, H.: Parameter variations in prediction skill optimization at ECMWF, Nonlinear Processes in Geophysics, 20, 1001–1010, https://doi.org/10.5194/npg-20-1001-2013, https://www.nonlin-processes-geophys.net/20/1001/2013/, 2013a.

Ollinaho, P., Laine, M., Solonen, A., Haario, H., and Järvinen, H.: NWP model forecast skill optimization via closure parameter variations, Quarterly Journal of the Royal Meteorological Society, 139, 1520–1532, https://doi.org/10.1002/qj.2044, https://rmets.onlinelibrary.wiley.com/doi/abs/10.1002/qj.2044, 2013b.